# Production of IFN-γ by splenic dendritic cells during innate immune responses against *Francisella tularensis* LVS depends on MyD88, but not TLR2, TLR4, or TLR9

**Roberto De Pascalis**[1]*, **Amy P. Rossi**[1¤], **Lara Mittereder**[1], **Kazuyo Takeda**[2], **Adovi Akue**[3], **Sherry L. Kurtz**[1], **Karen L. Elkins**[1]*

**1** Division of Bacterial, Parasitic and Allergenic Products, Laboratory of Mucosal Pathogens and Cellular Immunology, Center for Biologics Evaluation and Research, Food and Drug Administration, Silver Spring, Maryland, United States of America, **2** Microscopy and Imaging Core, Center for Biologics Evaluation and Research, Food and Drug Administration, Silver Spring, Maryland, United States of America, **3** Flow Cytometry Core, Center for Biologics Evaluation and Research, Food and Drug Administration, Silver Spring, Maryland, United States of America

¤ Current address: Immunology Graduate Program, University of Cincinnati College of Medicine, Cincinnati, Ohio, United States of America
* roberto.depascalis@fda.hhs.gov (RDP); karen.elkins@fda.hhs.gov (KLE)

## Abstract

Production of IFN-γ is a key innate immune mechanism that limits replication of intracellular bacteria such as *Francisella tularensis* (*Ft*) until adaptive immune responses develop. Previously, we demonstrated that the host cell types responsible for IFN-γ production in response to murine *Francisella* infection include not only natural killer (NK) and T cells, but also a variety of myeloid cells. However, production of IFN-γ by mouse dendritic cells (DC) is controversial. Here, we directly demonstrated substantial production of IFN-γ by DC, as well as hybrid NK-DC, from LVS-infected wild type C57BL/6 or Rag1 knockout mice. We demonstrated that the numbers of conventional DC producing IFN-γ increased progressively over the course of 8 days of LVS infection. In contrast, the numbers of conventional NK cells producing IFN-γ, which represented about 40% of non-B/T IFN-γ-producing cells, peaked at day 4 after LVS infection and declined thereafter. This pattern was similar to that of hybrid NK-DC. To further confirm IFN-γ production by infected cells, DC and neutrophils were sorted from naïve and LVS-infected mice and analyzed for gene expression. Quantification of LVS by PCR revealed the presence of *Ft* DNA not only in macrophages, but also in highly purified, IFN-γ producing DC and neutrophils. Finally, production of IFN-γ by infected DC was confirmed by immunohistochemistry and confocal microscopy. Notably, IFN-γ production patterns similar to those in wild type mice were observed in cells derived from LVS-infected TLR2, TLR4, and TLR2xTLR9 knockout (KO) mice, but not from MyD88 KO mice. Taken together, these studies demonstrate the pivotal roles of DC and MyD88 in IFN-γ production and in initiating innate immune responses to this intracellular bacterium.

**Data Availability Statement:** All relevant data are within the manuscript and its Supporting Information files.

**Funding:** The authors received no specific funding for this work.

**Competing interests:** The authors have declared that no competing interest exist.

## Introduction

Dendritic cells (DC) play a crucial role in the development of specific immune responses against infections. DC bridge innate and adaptive immune responses by processing and presenting antigen in the context of MHC Class I and/or II, by expressing T cell co-stimulatory molecules, and by producing cytokines. During innate immune responses, DC, neutrophils, and natural killer (NK) cells represent the first line of defense against infection, coordinating to contain microbial replication while adaptive immune responses develop. Through Toll-like receptor activation in response to pathogen-derived microbial products, DC and NK cells interact, resulting in NK activation and DC maturation [1]. In an *in vitro* model of *Chlamydia psittaci* infection, activation of NK cells and strong IFN-γ production may occur also by release of exosomes from infected DC [2].

Another mechanism of defense against intracellular bacteria including *Salmonella typhimurium*, *Listeria monocytogenes*, *Francisella tularensis* (*Ft*), and *Mycobacterium tuberculosis* is the production of IFN-inducible proteins such as AIM2 [3, 4]. This response mechanism is associated with increases in caspase-1, IL-1β, and IL-18 production by DC, which in turn induce IFN-γ production by T cells [5]. However, following infection with the attenuated vaccine strain of *Ft*, Live Vaccine Strain (LVS), IFN-γ can also be produced directly by CD11c$^+$ cells in spleens [6] suggesting that DC contribute to IFN-γ production in parallel with production by NK cells and before T cells.

Intracellular infection by *Legionella pneumophila*, *Salmonella typhimurium*, and *Yersinia enterocolitica*, or contact with bacterial antigens from *Pseudomonas aeruginosa*, *Streptococcus pneumoniae* or *Shigella flexneri*, can induce apoptotic programmed cell death of DC, which facilitates recruitment of additional inflammatory cells and control of bacterial replication [7]. In some circumstances, however, *Salmonella*-induced DC death may be associated with reduced antigen presentation; further, intra-dendritic cell infection can be used by *Salmonella* as transport for spreading [8]. In contrast, *Listeria monocytogenes* infection of DC does not induce apoptosis, and DC survive while maintaining their ability to process bacteria and to present antigens [9, 10].

In other circumstances, *Salmonella*, *Yersinia*, or *Helicobacter* use different strategies to evade intestinal DC recognition, and therefore limit T cell activation [11]. *Mycobacteria*-infected DC that are outside granulomas migrate less efficiently than non-infected DC. This results in reduced antigen availability in lymph nodes, and therefore in reduced T-cell response [12]. Moreover, *Mycobacteria* ligands can activate immunosuppressive pathways, leading to suppression of DC maturation and antigen presentation [13, 14]. These examples indicate that the immune responses mediated by DC vary depending on the intracellular bacteria involved, and different subsets of DC may be involved in this variability.

*Ft* subsp. *tularensis* causes severe disease in animals and occasionally in humans after exposure to low numbers of bacteria by several routes, including respiratory exposure. Following inhalation of bacteria, lung DC and alveolar macrophages are targeted by *Ft* for invasion and replication. *Ft* deploys several effective evasion strategies to counteract host immune defenses in both the extracellular space and in intracellular compartments [15–17]. In addition, virulent *Ft* can actively suppress pro-inflammatory cytokine responses by human monocytes [18]. Lack or low abundance of immune factors such as CD14 in lung DC may further contribute to evasion of innate immune responses [19]. Moreover, infection of mice with a lethal dose of virulent *Ft* can induce expression of pro-inflammatory cytokine mRNA in livers within 48 hours, but this was insufficient to prevent lethality [20].

*Ft* LVS, which is derived from *Ft* subsp. *holoarctica*, is attenuated in humans, but its virulence in mice varies with the route of infection [21]. Aerosol and intranasal LVS infection

models have highlighted the role of pro-inflammatory cytokines, including IFN-γ, during induction of protection [22]. LVS is therefore used in murine models to evaluate immune responses against *Ft* and other intracellular pathogens. Similar to virulent *Ft*, LVS can infect DC *in vitro* and *in vivo* [23, 24]. Although reports on the nature of the LVS-DC interactions and DC cytokine production are conflicting [25], it appears that LVS-infected DC, in contrast to infection of DC with fully virulent *Ft*, initiate effective immune responses against bacteria. These interactions are modulated by TLR2 and MyD88 signaling, but not by TLR4 engagement, after intranasal or intradermal LVS infection [26]. However, the same signaling may induce priming when activated by LVS [27] or tolerance when activated by the virulent *Ft* SchuS4 [28]. The contributions of DC cytokine production in general and IFN-γ production in particular to successful vaccination against *Ft* therefore remain incompletely understood. In this study, we used a variety of analytical methods to definitively demonstrate that conventional DC are actively involved in innate immune responses against LVS by producing IFN-γ, even when infected themselves. Further, DC IFN-γ production is dependent on MyD88 but not on TLR2, TLR4, or TLR9.

## Materials and methods

### Experimental animals

Six- to twelve-week-old specific-pathogen-free male C57BL/6J and B6.129S7-Rag1<tm1Mom>/J KO (Rag1 KO) mice were purchased from Jackson Laboratories (Bar Harbor, Maine); breeding pairs of TLR2 KO, TLR4 KO, and MyD88 were purchased from Jackson Laboratories, and TLR/MyD88 KO mice as well as TLR2x9 KO mice were bred in house. All mice were housed in sterile microisolator cages in a barrier environment at CBER/FDA, fed autoclaved food and water *ad libitum*, and routinely tested for common murine pathogens by a diagnostic service provided by the Division of Veterinary Services, CBER. Within an experiment, all mice were age matched. At selected time points or at end of a study, animals were euthanized with carbon dioxide inhalation in a euthanasia chamber where carbon dioxide was introduced at the rate of at least 20% of the chamber volume per minute. No animals were subjected to anesthesia.

### Ethics statement

All experiments were performed under protocols approved by the Animal Care and Use committee of CBER. These protocols meet the standards for humane animal care and use set by the Guide for the Care and Use of Laboratory Animals and PHS policy.

### Bacteria and growth conditions

*F. tularensis* LVS (American Type Culture Collection 29684) was grown to mid-log phase in modified Mueller-Hinton (MH) broth (Difco Laboratories, Detroit, MI), as previously described [29, 30], harvested, and frozen in aliquots in broth alone at -80ºC. LVS expressing GFP was prepared using the GFP expressing plasmid pKK214 [31]. To generate GFP-LVS, LVS grown in MH broth was pelleted by centrifugation, washed three times with 0.5 M sucrose, and resuspended in ~200 μl 0.5 ml sucrose. One hundred μl of diluted LVS was combined with 100 ng pKK214 and transformed by electrophoresis using a Gene Pulser Electroporator II (BioRad, Hercules, CA), using standard procedures. A second sample of diluted LVS, but without plasmid DNA, served as a control for spontaneous antibiotic resistance. Samples were transferred into MH broth for 2 hours at 37˚C, and cultures were then plated onto MH agar containing 10 μg/ml kanamycin for selection of plasmid transformants. GFP florescence of transformants was confirmed by visualization of agar plates with a LAS-3000 imaging

system (Fujifilm Medical Systems, Stamford, CT). Selected clones were grown to mid-log phase in MH broth containing 10 μg/ml kanamycin and frozen at -80˚C. Each clone was tested for virulence by assessment of growth in murine bone marrow-derived macrophages, and all exhibited growth characteristics similar to the original parental LVS strain. One clone was selected for use in subsequent studies. LVS expressing mCherry was previously described [32] and also exhibited growth and virulence similar to LVS.

## Bacterial infections

For each independent experiment, 3–5 mice were immunized by intradermal (i.d.) injection with 1 x $10^5$ colony forming units (CFU) LVS or GFP-labeled LVS, diluted in 0.1 ml phosphate-buffered saline (PBS) (BioWhittaker/Lonza, Walkersville, MD), and euthanized for analyses on the indicated days. Actual doses of inoculated bacteria were simultaneously determined by retrospective plate count; control groups received 0.1 ml PBS i.d.

## Determination of bacterial organ burdens

To determine the number of CFU in spleens of infected mice, organs were removed aseptically and disrupted with a 3-ml syringe plunger in 10 ml of sterile PBS/2% fetal bovine serum (FBS). Appropriate dilutions were plated on MH plates. After 2–3 days incubation at 37ºC/5% $CO_2$, bacterial CFU were counted and the results calculated according the dilution factors used.

## Preparation of splenocytes

Isolated spleens were used to prepare single-cell suspensions; erythrocytes were lysed with ammonium chloride (ACK lysing buffer, BioWhittaker/Lonza). Cells were washed with 5 mL EDTA buffer, viability was assessed by exclusion of trypan blue, and cell concentrations were adjusted as required. In selected experiments, to improve the yield of dendritic cells isolated spleens were initially treated with 5 ml of digestion buffer, composed of RPMI/2mM EDTA and containing collagenase (1gr/L) and DNase I (120 IU/μl), minced, and incubated for 30 minutes at 37˚C in a 5% $CO_2$ incubator before washing.

## *In vitro* cell purification

A total of 3–4 x $10^8$ splenocytes were used to deplete B and T cell subpopulations by the Dynabeads[TM] system (Invitrogen, Carlsbad, CA) using Biotin Binder magnetic beads (Invitrogen), according to the manufacturer's instructions. Biotin anti-mouse CD19 and biotin anti-mouse TCRβ were used to deplete B and T cells, respectively. The Dynal beads system was also used to enrich dendritic cells. The composition and relative purity of the resulting depleted and enriched cells were assessed by multiparameter flow cytometry.

## *In vivo* depletion of natural killer (NK) cells

To selectively deplete NK cells, mice were injected intraperitoneally (i.p.) twice with 200 μg of anti-NK1.1 monoclonal antibody (PK136) produced by BioXCell (West Lebanon, NH), as previously described [33].

## Flow cytometry

Single cell suspensions were prepared from total and depleted splenocytes in EDTA buffer and incubated with 10 μl anti-CD16/CD32 (Fc block, BD Pharmingen, San Diego, CA). To discriminate live from dead cells, a staining step was performed using a commercially available kit and following manufacturers' instructions (Live/dead staining kit, Invitrogen, Carlsbad,

CA). The cells were then washed in flow cytometry buffer (PBS with 2% FBS) and stained for cell surface markers. Antibody concentrations were previously optimized for use in multicolor staining protocols as required, using appropriate fluorochrome-labeled isotype matched control antibodies. The following antibodies were used: anti-CD19 (clone 1D3), anti-TCRβ (clone H57-597), anti-CD45 (clone 30-F11), anti-NK1.1 (clone PK136), anti-CD11b (clone M1/70), anti-CD11c (cloneHL3), anti-Ly6C (clone AL-21) anti Ly6G (clone 1A8) and anti-I-A/I-E (cloneM5/114.15.2); all antibodies were purchased from BD Pharmingen or Biolegend (San Diego, CA). For intracellular staining (ICS) of IFN-γ, splenocytes were initially incubated with brefeldin A (Sigma-Aldrich, St. Louis, MO) for 4 hours, then washed and stained for surface markers. Cells were then fixed for 20 minutes at room temperature with 2% paraformaldehyde (EMS, Hatfield, PA), washed and permeabilized with Perm/Wash buffer (BD Pharmigen), and then stained with anti-IFN-γ (clone XMG1.2) (BD, Pharmingen). After 30 minutes incubation, cells were washed in Perm/Wash buffer and fixed in 0.5% paraformaldehyde. Ten—thirty thousand total events were collected using an analytical LSR II or LSR Fortessa flow cytometer (Becton Dickinson). Data analyses were performed using FlowJo (Tree Star, Inc.) software as previously described [6, 34].

## Cell sorting

To sort cells, $5x10^6$–$1.5x10^7$ splenocytes depleted of B and T cells were resuspended in sorting buffer. Aggregates were then gated out using SSC-W / SSC-H and FSC-W / FSC-H parameters, and live CD45[+] gating was used to gate out debris and dead cells. Conventional DC were defined and sorted by positivity of both CD11c and I-A/I-E markers. CD11c[-] I-A/I-E[-] CD11b[+] Ly6G[+] cells were defined and sorted as neutrophils. Conventional natural killer cells were defined as CD11c[-] I-A/I-E[-] CD11b[-] NK1.1[+], and monocytes were identified as CD11c[-] I-A/I-E[-] CD11b[+] Ly6C[+] Ly6G[-]. Fluorescence minus one (FMO) controls were used to identify and gate cell populations. Approximately 0.3–4 $x10^6$ cell subpopulations were sorted and used for real-time PCR.

## Real time PCR

Genomic DNA and total RNA were simultaneously purified from sorted single-cell suspensions using a commercial kit (AllPrep DNA/RNA Mini kit; Qiagen, Valencia, CA), according to the manufacturer's instructions. 50–500 nanograms of RNA were used to synthesize cDNA using the commercially available High Capacity RNA-to-cDNA kit (Applied Biosystems, Carlsbad, CA), according to the manufacturer's instructions. Selected genes were amplified by semi-quantitative real-time PCR with a ViiA 7 sequence detection system (Applied Biosystems). In all qRT-PCR analyses, GUSB was used to normalize data; delta Ct (ΔCt) and the ratios between ΔCt of vaccine samples and control naïve samples were then calculated. Approximately 100 ng of purified DNA, which included mouse and bacterial DNA, was used to amplify the *Francisella* genes Tul4 and 23kDa [35]; genomic GAPDH served to estimate the amount of eukaryotic DNA. Absolute quantifications of Tul4 and 23kDA were calculated using corresponding standard curves. To generate standard curves, DNA was prepared from LVS, the concentration was calculated, and serial dilutions were prepared. Bacteria were quantified using genomic equivalents (GE), where 1 ng DNA = 489,000 GE and 1 GE = 1 bacterium [35, 36]. Averages of the absolute quantification of Tul4 and 23kDa were then calculated.

## *In vitro* cell cultures and protein quantitation by ELISA

To assess IFN-γ production *in vitro*, $5x10^6$ splenocytes, derived from naïve and LVS-infected mice, were cultured in 24 well plates for three days in 1 ml/well DMEM complete media

(BioWhittaker), defined as DMEM containing 10% fetal bovine serum (HyClone, Logan, UT), 1% glutamine, 1% HEPES, 1% sodium bicarbonate, 1% sodium pyruvate, and 1% non-essential amino acids (all from BioWhittaker/Lonza). Culture supernatants were assayed for IFN-γ using a standard sandwich ELISA, according to the manufacturer's instructions (BD Pharmingen). IFN-γ was quantitated by comparison to recombinant standard protein using four-parameter fit regression in the SOFTMax Pro ELISA analysis software (Molecular Devices, San Jose, CA).

## Immunohistochemistry

Spleens from C57BL/6 or Rag KO mice, naïve or infected with GFP-LVS were prepared for cryosections that were used to perform immunostaining. Rabbit anti-GFP polyclonal antibody (ThermoFisher, Waltham, MA), purified rat anti-mouse IFN-γ (Biolegend, clone XMG1.2), and purified Armenian hamster anti-mouse CD11c (Biolegend, clone N418) antibodies were used as primary antibodies. Alexa 488 donkey anti-rabbit IgG, Alexa 594 donkey anti-rat IgG and Alexa 647 goat anti-Armenian hamster IgG (Jackson Immunoresearch, West Grove, PA) were used for detection, respectively. Counterstaining was performed using Hoechst 33258 (Thermofisher, Walthan MA). Stained slides were scanned by a NanoZoomer XR (Hamamatsu corporation, Japan) and images were stored as ndpi format. Images were imported and analyzed using Imaris (Bitplane, Concord MA).

## Intracellular staining of sorted CD11c cells and confocal microscopy

Splenocytes from C57BL/6 or Rag1 KO mice, naive or infected with mCherry-LVS or GFP-LVS, were prepared as described above. Briefly, surface staining was performed using either FITC- or Alexa 594-conjugated anti-mouse CD11c (clone N418, Biolegend). For intracellular staining, cells were then fixed, treated with permeabilization buffer and followed by staining with Alexa 647-conjugated anti-mouse IFNγ (clone XMG1.2, Biolegend). Nuclei were counterstained either with Hoechst 33258 or DAPI (ThermoFisher). Images were acquired using a TCS SP8 confocal microscope (Leica Microsystems, Germany), and images were stored as lif format. Huygens professional (Scientific Volume Imaging, Netherland) and Imaris (Bitplane) were used for image analyses.

## Statistical analyses

Microsoft Excel was used to evaluate differential bacterial growth, IFN-γ production, and gene expression. CFU data were $log_{10}$ transformed and cytokine concentrations were measured using a log scale; thus, a normal distribution was assumed. Significant differences were evaluated using a two-tailed Student's *t* test, with a *P* value of $< 0.05$ indicating significance. Corrections for multiple comparisons were performed using the Bonferroni method.

# Results

## DC subpopulations in LVS-infected mice produce substantial amounts of IFN-γ

To visualize the presence of CD11c$^+$ dendritic cells (DC) within infected spleens, Rag1 KO mice, which do not have mature B or T cells, were infected with GFP-LVS for four days, and spleens were removed and stained. Scattered CD11c$^+$ cells and small numbers of IFN-γ$^+$ cells were seen in naïve mice (Fig 1A), but the numbers of both increased markedly after infection (Fig 1B). IFN-γ appeared to be present in both CD11c$^+$ and CD11c$^-$ cells of infected spleens (Fig 1B and insert). GFP$^+$ LVS bacteria were detected throughout the spleens of infected mice, and bacteria were in proximity to some of the IFN-γ-producing cells.

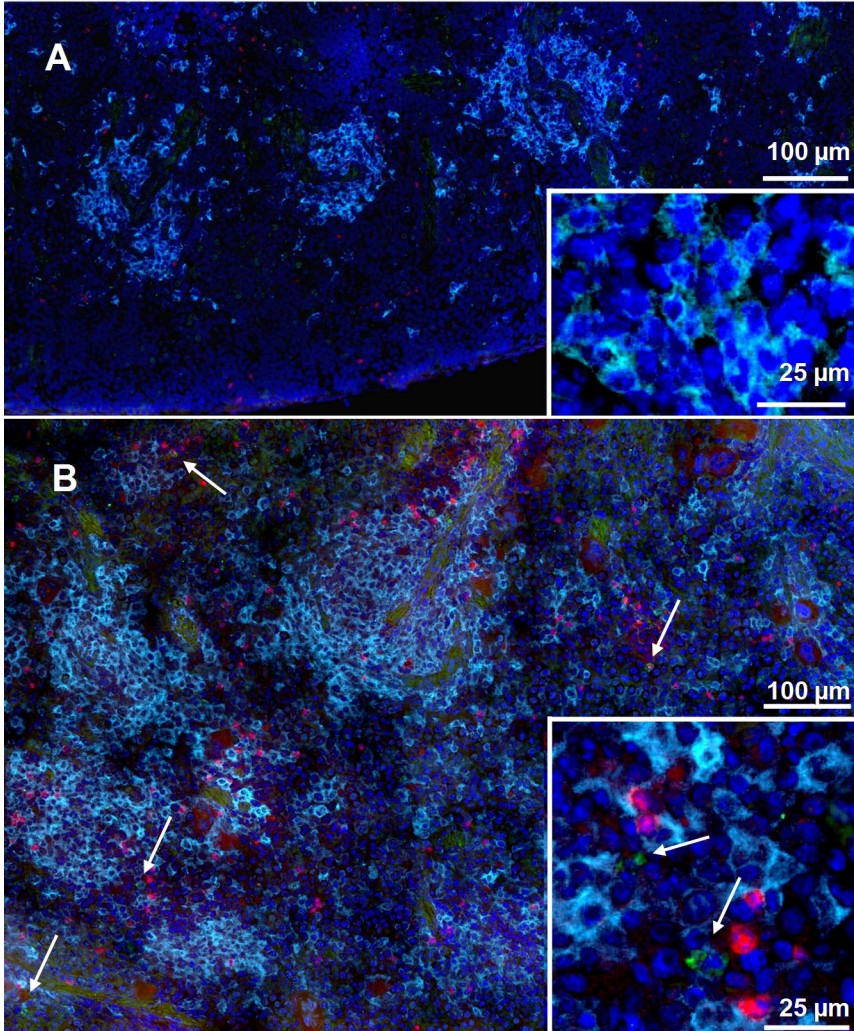

**Fig 1. IFN-γ-producing dendritic cells increase in spleens upon *Ft* LVS infection.** Rag1 KO mice were infected with $10^5$ GFP-LVS for four days. Spleens from non-infected (A) and GFP-LVS infected (B) mice were sectioned and prepared for immunohistochemistry staining. Anti-mouse IFN-γ (red) and anti-mouse CD11c (cyan) were used as primary antibodies; nuclei stained with DAPI appear dark purple. An anti-GFP polyclonal antibody (green) was used to enhance the signal from LVS. Arrows indicate GFP-LVS. Images are representative of those evaluated in three mice.

To identify dendritic cell subpopulations responsible for IFN-γ production, splenocytes from naïve and LVS infected C57BL/6 or Rag1 KO mice were stained for surface makers and intracellular IFN-γ (ICS) and analyzed by flow cytometry. Earlier studies showed that the number and distribution of IFN-γ-producing myeloid cells were comparable between splenocytes from LVS-infected C57BL/6 and Rag1 KO mice [6]. Here, in addition to the CD11c marker, MHCII was used to discriminate NK cells from DC subpopulations. Conventional myeloid DC (cDC; MHCII$^+$ CD11c$^+$ NK1.1$^-$ B220$^-$ Gr1$^-$ CD11b$^+$) as well as other CD11c$^+$ cells produced IFN-γ (Fig 2). Some of these cells were also NK1.1$^+$ and are referred to here as NK-DC (CD11c$^+$ NK1.1$^+$ MHCII$^{+/-}$ Gr1$^-$ CD11b$^{+/-}$). Conventional NK cells (NK1.1$^+$ MHCII$^-$ CD11c$^-$) represented about 40% of non-B/T IFN-γ-producing cells, while conventional DC and NK-DC averaged about 30% each (Figs 2 and 3). Time course studies demonstrated that numbers of IFN-γ-producing NK cells (Fig 3A) and NK-DC (Fig 3B) peaked at day 4 after

# TCRβ⁻ CD19⁻ IFN-γ-producing cells

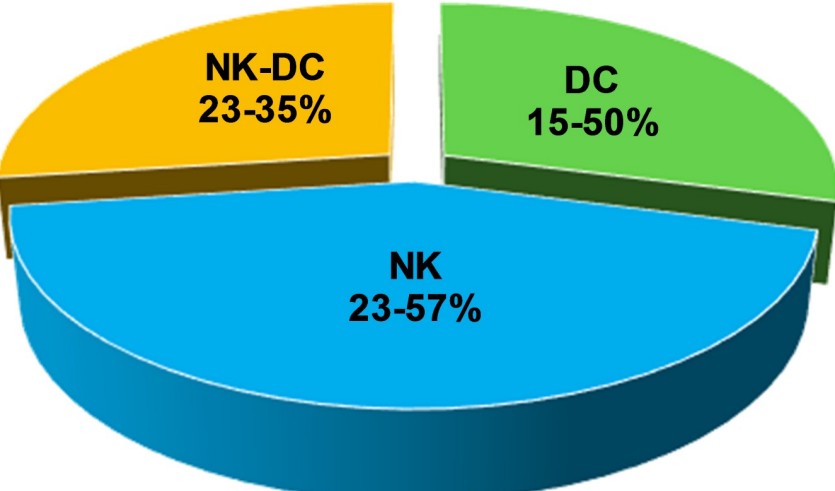

**Fig 2. Multiple myeloid subpopulations produce IFN-γ during innate immune response against *Ft* LVS infection.**
C57BL/6 mice were infected with $10^5$ LVS. After four days, single cell suspensions of splenocytes from LVS-infected were prepared and analyzed by flow cytometry. After exclusion of fragments, aggregates, and dead cells, IFN-γ producing non-B non-T cells were quantified according to surface markers coupled with ICS. DC were identified as $CD11c^+$ $MHCII^+$ $NK1.1^-$ $B220^-$ $CD11b^+$ $Gr1^-$. NK cells were identified as $NK1.1^+$ $CD11c^-$ $MHCII^-$. NK-DC cells were identified as $NK1.1^+$ $CD11c^+$ $MHCII^{+/-}$ $CD11b^{+/-}$ $Gr1^-$. The relative proportion of IFN-γ production by each cell type is shown using a pie chart. Data represent the range of proportions found in five independent experiments of similar design and outcome.

LVS infection. In contrast, the production of IFN-γ by cDC progressively increased during the first week of infection (Fig 3C).

To further confirm the role of conventional DC in producing IFN-γ, we evaluated IFN-γ production by myeloid cells that were depleted of NK cells and highly enriched for $CD11c^+$ cells. Rag1 KO mice were depleted *in vivo* of $NK1.1^+$ cells before LVS infection; four days later, splenocytes derived from these mice were then enriched *in vitro* for $CD11c^+$ cells using magnetic beads (S1 Fig). The resulting cells were cultured for 3 days, without further stimulation, and supernatants obtained from these cultures were analyzed for IFN-γ production (Fig 4A). Levels of IFN-γ in supernatants from the $CD11c^+$-enriched cells were quite substantial. For context, the amounts of IFN-γ produced by total splenocytes from Rag 1 KO mice under the same conditions are shown (Fig 4B). The enrichment studies confirm that DC produce a substantial proportion of the IFN-γ in spleens found within a few days after murine LVS infection.

## DC are directly infected with LVS *in vivo*

To directly evaluate the status of LVS infection in DC, splenocytes derived from naïve and mCherry-LVS-infected C57BL/6 mice were depleted of B and T cells using magnetic beads. Following the depletion of B and T cells, the remaining myeloid splenocytes from naïve and infected mice were enriched for DC. The resulting cells were stained and analyzed by confocal microscopy (Fig 5). In contrast to naïve myeloid cells (Fig 5A and 5B), mCherry-LVS was present in $CD11c^+$ cells as well as $CD11c^-$ cells from LVS-infected mice (Fig 5C and 5D). In a few cases, colocalization of $CD11c^+$ and mCherry was readily visible (Fig 5C), but in other examples the cell morphology was disrupted, suggesting deteriorating health of the cells following infection.

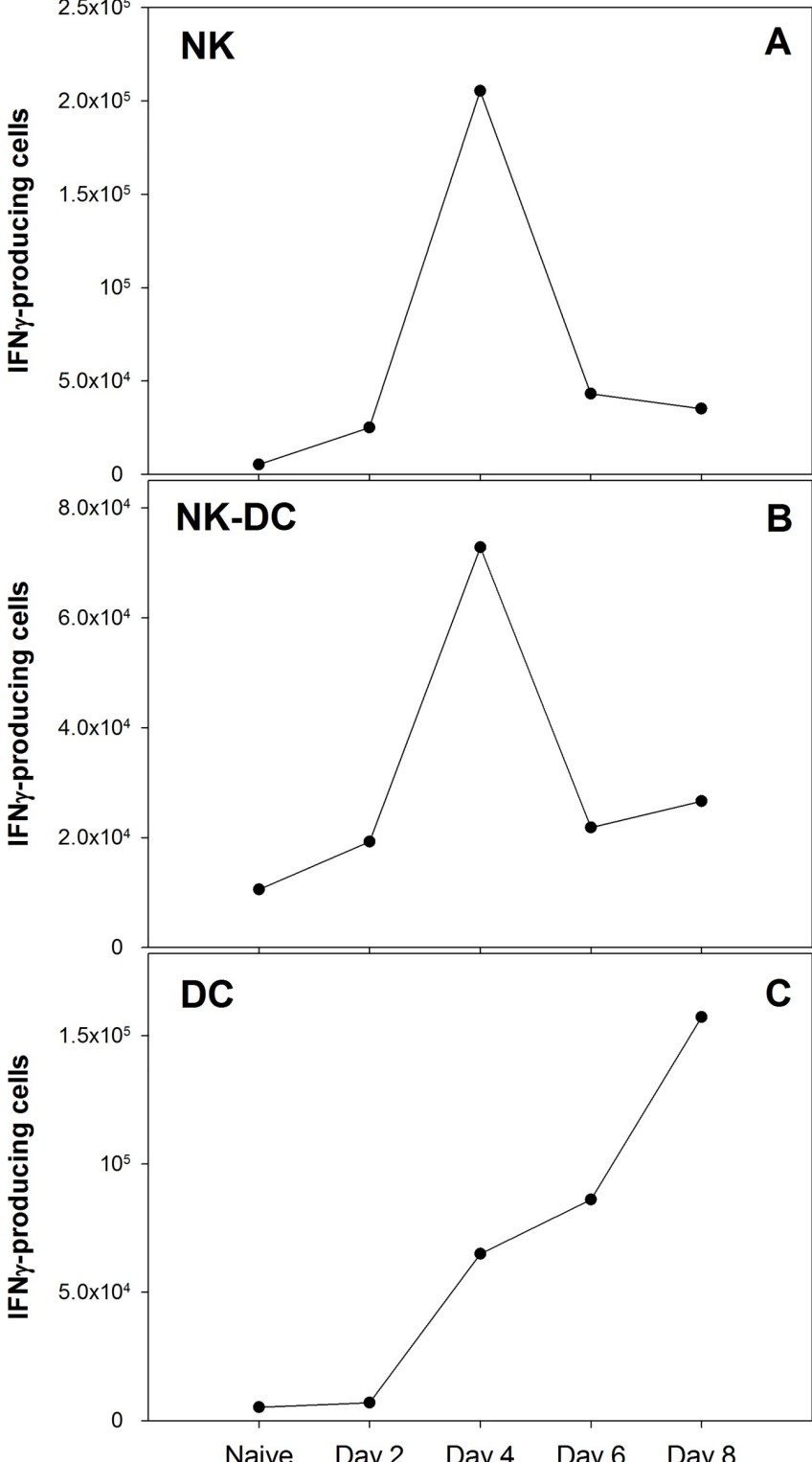

**Fig 3. IFN-γ production by DC and NK subpopulations varies over time after *Ft* LVS infection.** C57BL/6 mice were infected with $10^5$ LVS. Single cell suspensions of splenocytes from LVS-infected C57BL/6 mice were prepared and analyzed by flow cytometry at days 2, 4, 6, and 8 after infection. Splenocytes from non-infected mice served as a negative control (naïve). After exclusion of fragments, aggregates, and dead cells, IFN-γ producing non-B non-T cells were quantified according to surface markers coupled with ICS. DC were identified as CD11c$^+$ MHCII$^+$ NK1.1$^-$ B220$^-$

CD11b⁺ Gr1⁻. NK were identified as CD11c⁻ MHCII⁻ NK1.1⁺. NK-DC cells were identified as CD11c⁺ MHCII^{+/-} NK1.1⁺ CD11b^{+/-} Gr1⁻. Each dot represents total cell counts derived from three pooled spleens. Data are from one complete time course experiment.

To further confirm the role of DC and other myeloid cells in controlling LVS during early stages of infection, splenocytes from naïve and LVS-infected WT mice were depleted of B and T cells and sorted by flow cytometry to purify total DC (live CD45⁺ CD11c⁺ MHCII⁺ cells) and neutrophils (live CD11c⁻ MHCII⁻ CD11b⁺ Ly6G⁺ cells; gating strategy shown in S2 Fig). DNA from sorted DC and neutrophils was used to amplify *Francisella* Tul4 and 23kDA genes [35], and amplicons were quantified using standard curves prepared with LVS DNA. Cytokine production was determined from mRNA and quantified in relationship to naïve cells. Bacterial DNA was readily detected in sorted DC and neutrophils, suggesting that these cells were either infected by LVS *in vivo* or LVS was tightly adhered to cells (Table 1, DNA). Analyses of gene expression in sorted cells from LVS-infected mice indicated substantial upregulation of IFN-γ and IL-1α mRNA in DC, as well as in neutrophils compared to cells from naïve mice (Table 1, RNA). In contrast, TNF-α and IL-12 p40 were less upregulated.

Finally, to evaluate whether individual IFN-γ-producing DC were simultaneously infected with LVS, splenocytes from GFP-LVS-infected Rag1 KO mice were analyzed by confocal microscopy (Fig 6). CD11c expression clearly identified DC from both naïve (Fig 6A) and LVS-infected mice (Fig 6B), and GFP and IFN-γ signals were detected only in cells from LVS-

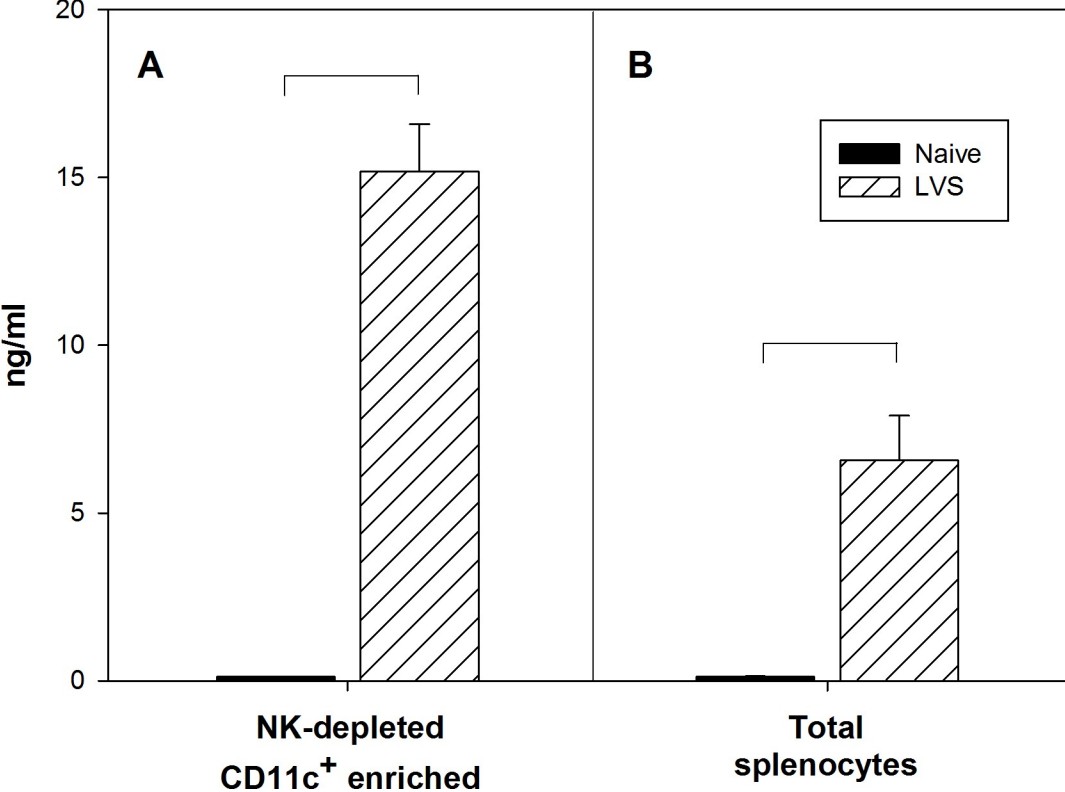

**Fig 4. DC produce high amount of IFN-γ during innate immune responses to *Ft* LVS.** Rag1 KO mice were depleted *in vivo* of NK cells and infected with 10⁵ LVS i.d. Splenocytes were enriched *in vitro* for CD11c⁺ cells using magnetic beads and cultured for three days. Supernatants were then collected and analyzed for IFN-γ production by ELISA (panel A). Production of IFN-γ from total splenocytes is shown for context (panel B). Data are from one experiment.

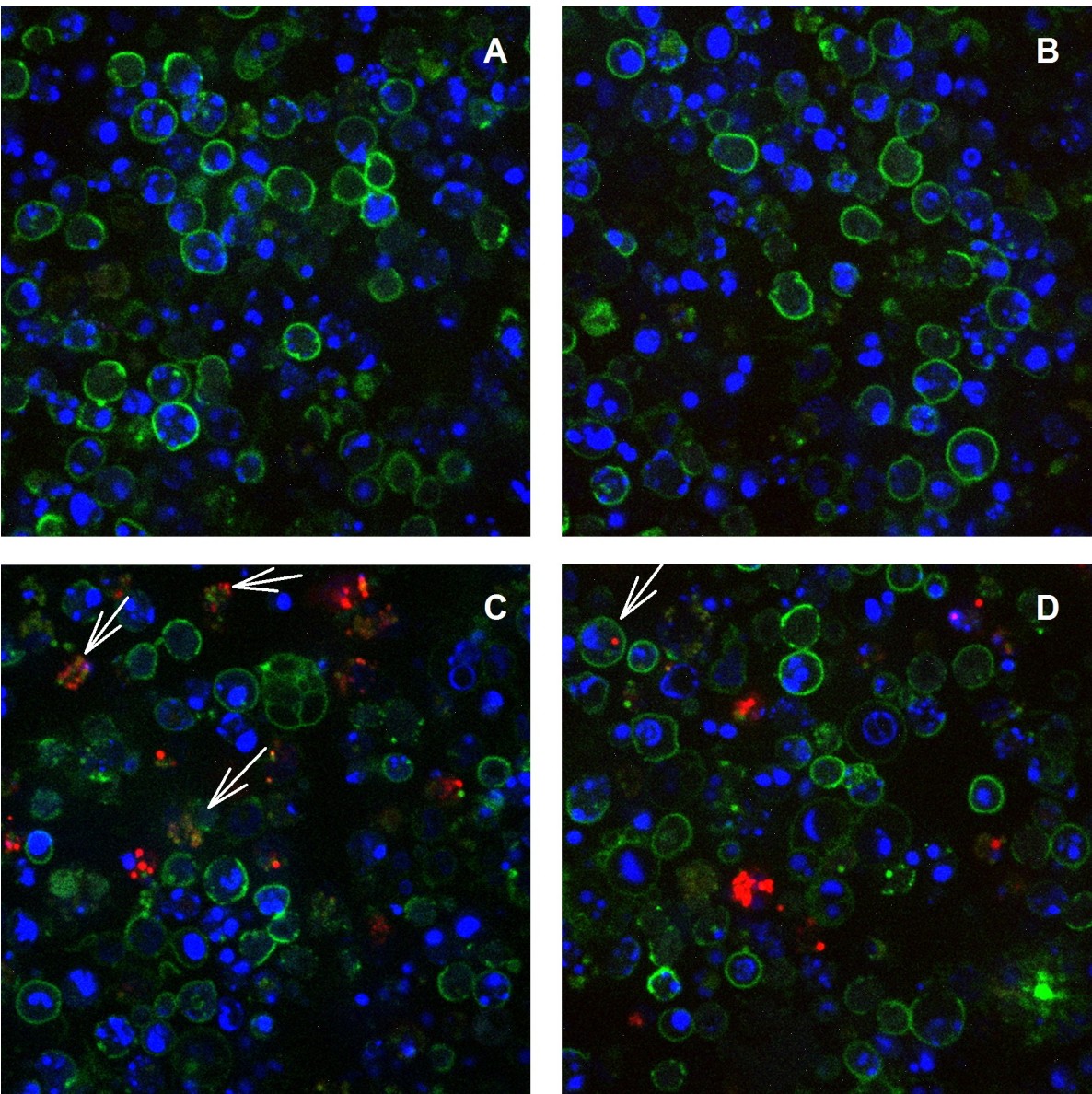

**Fig 5. DC are infected *in vivo* by *Ft* LVS.** C57BL/6 mice were infected with 10⁵ LVS i.d. Four days after infection, single cell suspensions of splenocytes from naïve (panels A and B) and mCherry-LVS infected mice (panels C and D) were depleted of B and T cells using magnetic beads, and the resulting cells were analyzed by confocal microscopy. CD11c⁺ cells (green) were identified according to the surface marker. LVS-mCherry was identified as red, and all cells were counterstained with DAPI (blue) to identify nuclei. Arrows indicate mCherry-LVS-infected CD11c⁺ cells. Images are representative of those evaluated in two independent experiments of similar design and outcome.

infected mice. Most importantly, imaging analyses identified multiple DC that were infected with LVS and that also expressed IFN-γ (Fig 6B).

## MyD88 deficiency, but not TLR2/4/9 deficiencies, increases LVS bacterial burden and decreases IFN-γ production

To evaluate the role of TLRs in modulating IFN-γ production by splenocytes subpopulations, particularly DC, TLR2 KO, TLR4 KO, TLR2x9 KO, and MyD88 KO mice were infected with

**Table 1. LVS infects both DC and neutrophils *in vivo*, but cytokine gene expression varies between cell types.**

| LVS-infected C57BL/6 | | | |
|---|---|---|---|
| Dendritic cells | DNA | Tul4 | 8,001[a] |
| | | 23kDa | 5,324 |
| | RNA | IFN-γ | 330[b] |
| | | IL-1α | 22.1 |
| | | IL-12β | 1.6 |
| | | TNF-α | 2.4 |
| Neutrophils | DNA | Tul4 | 4,573 |
| | | 23kDa | 7,577 |
| | RNA | IFN-γ | 32.4 |
| | | IL-1α | 127 |
| | | IL-12β | 1.07 |
| | | TNF-α | 10.6 |

DNA from cells sorted to purify DC or neutrophils (see Materials and Methods, and S2 Fig) was amplified using *Francisella*-specific primers for Tul4 and 23kDA genes. Absolute quantifications were calculated using corresponding standard curves, using serial dilutions of DNA prepared from LVS. Bacteria were quantified using genomic equivalents (GE) where 1 ng DNA = 489,000 GE and 1 GE = 1 bacterium.

[a] DNA data indicate GE per ~100 ng of total mouse and bacterial DNA. The upper limit of detection was $1 \times 10^8$ GE; the lower limit of detection was $1 \times 10^2$ and $1 \times 10^1$ GE for Tul4 and 23kDa, respectively. Bacterial DNA was undetectable in cells from uninfected mice. DNA data represent average of two independent experiments. mRNA from sorted cells was purified and analyzed for gene expression by RT-PCR.

[b] RNA data indicate median fold change compared to cells from uninfected mice, calculated from six independent experiments with sorted DC and five with neutrophils.

LVS. Organ burdens were determined on day 4, and cytokine gene expression and protein production in spleens were analyzed by RT-PCR and ELISA, respectively. As previously reported [37], MyD88 KO mice did not control LVS infection; further, using this dose and route of infection, LVS control was only modestly dependent on TLR2 and TLR9 (Fig 7A). Splenocytes from infected mice were cultured and the production of IFN-γ was evaluated by ELISA. While splenocytes from MyD88 KO mice produced little IFN-γ, those from C57BL/6J TLR2 KO, and TLR2/9 KO mice produced abundant and similar amounts of IFN-γ (Fig 7B). The observed differences in IFN-γ production were further supported by additional studies demonstrating abundant IFN-γ gene expression by C57BL/6J and all TLR KO mice (S3 Fig).

Finally, to evaluate the role of MyD88 and TLR2 in controlling LVS infection and specifically IFN-γ production by DC, TLR2 KO and MyD88 KO mice were infected with LVS, DC and neutrophils from these mice were sorted, and DNA and RNA were analyzed. In comparison to C57BL/6 mice (Table 1), the rate of LVS amplicons were about 4-fold higher in cells from TLR2 KO mice, but about 400-fold higher in DC and 3000-fold higher in neutrophils from MyD88 KO mice (Table 2). This is consistent with the much larger bacterial organ burdens in MyD88 KO mice (Fig 7A). Despite high bacterial burdens, however, the expression of IFN-γ in DC from MyD88 KO mice was quite low, unlike expression levels in TLR 2 KO mice. Unlike IFN-γ, cells from LVS-infected MyD88 KO mice produced similar amounts of IL-1α, IL-12β (IL-12p40), and TNF-α as those in cells from TLR2 KO mice. Collectively, these data demonstrate that the production of IFN-γ by DC after LVS infection depended on MyD88, but not TLR2, 4, or 9.

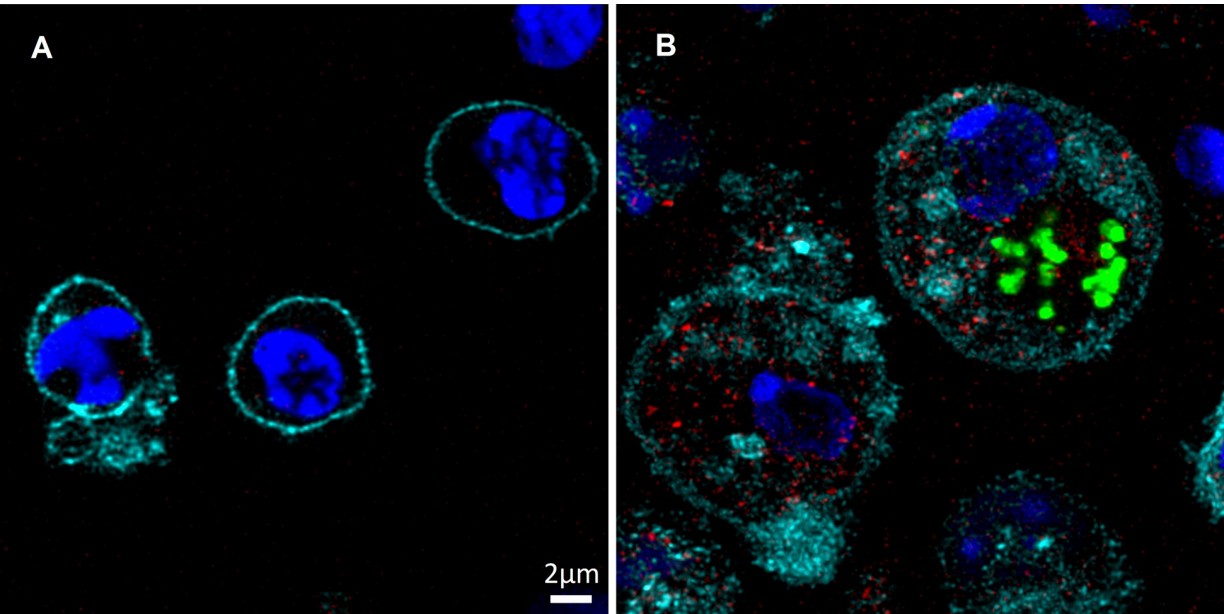

**Fig 6.** *Ft* **LVS-infected DC produce IFN-γ.** Rag1 KO mice were infected with $10^5$ LVS i.d. Four days after infection, single cell suspensions of splenocytes from naïve (panel A) and GFP-LVS-infected Rag1 KO mice (panel B) were prepared and analyzed by confocal microscopy. IFN-γ-producing (red) CD11c$^+$ cells (cyan) were identified by cell surface staining coupled with ICS. GFP-LVS was identified as green, and all cells were counterstained with DAPI (blue) to identify nuclei. Images are representative of those evaluated in two independent experiments.

## Discussion

We previously demonstrated that in addition to natural killer cells, other myeloid and lymphoid cell subpopulations produce IFN-γ during innate immune responses against infection with a prototypical intracellular bacterium, *Francisella* LVS [6]. Surprisingly, CD11c$^+$ cells from LVS-infected mice, whether NK1.1$^+$ or NK1.1$^-$, appeared to produce large amounts of this critical cytokine. In this study, we used different analytical methods, including flow cytometry, immunohistochemistry, and confocal microscopy, to definitively demonstrate that CD11c$^+$ DC produce IFN-γ following LVS infection, even when infected themselves. The substantial increase in DC numbers and wide distribution throughout infected spleens [38] suggests that these cells are highly involved in the earliest phases of innate immune responses against *Ft* infection. Therefore, DC may shape immune response outcomes by activities that are well beyond their traditional role in antigen presentation.

The present studies are consistent with previous reports demonstrating that both human and mouse DC, including CD8α$^+$ or CD8$^-$ DC, are substantial sources of IFN-γ production [39, 40]. Here, tracking CD8a expression on DC was precluded by the need to use the CD8 marker for T cell staining and the available instruments. DC cytokine production activities appear to depend on T-bet [41] and are subject to exhaustion as DC mature [42]. Production of IFN-γ by mouse DC has been reported in response to other intracellular infections such as *Burkholderia mallei* [43], as well as malaria [44], fungal [45], or parasitic [46] infection. Similarly, human monocyte-derived DC produce IFN-γ following stimulation with *Salmonella typhimurium* [47] and *M. bovis* BCG [48].

Although CD11c has been typically used as a marker for DC, the presence of this marker in other lineages makes the identification of DC subpopulations problematic. We therefore used MHCII to definitively discriminate DC from traditional NK cells (MHCII$^-$), which represented about 40% of IFN-γ-producing cells in the spleens of LVS-infected mice at day 4 after infection

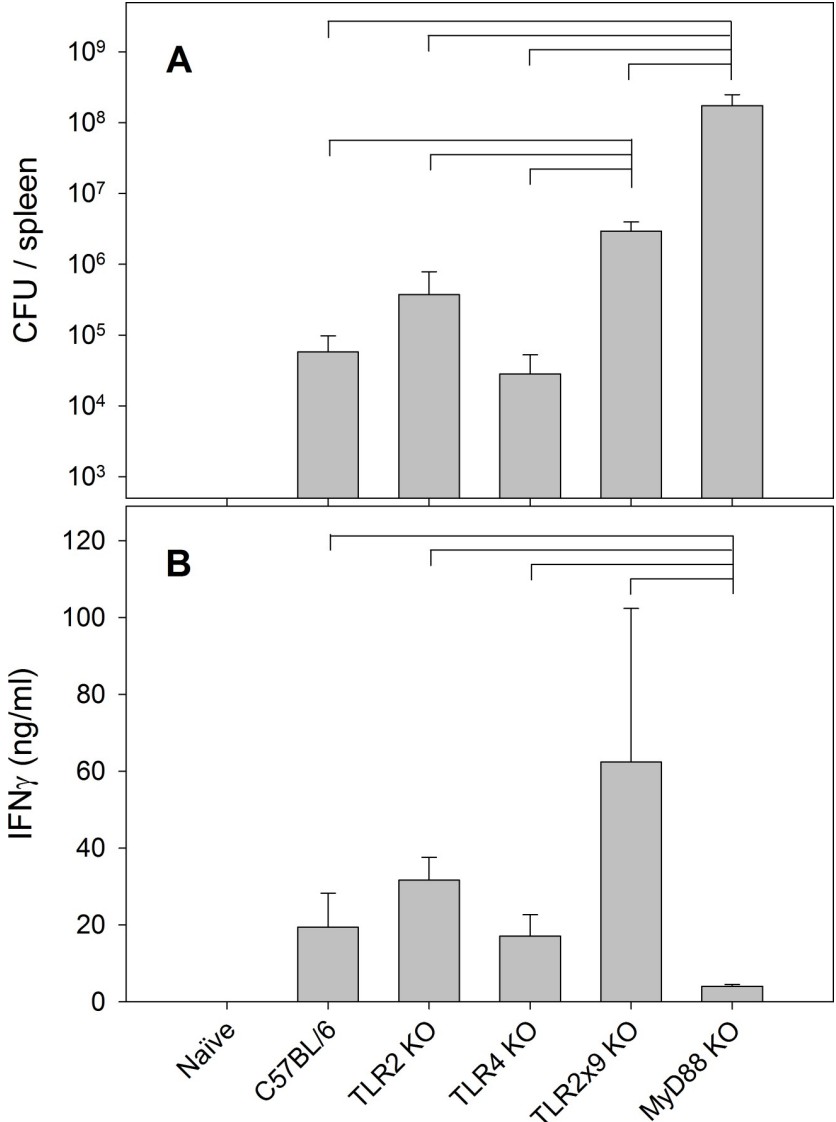

**Fig 7. IFN-γ production by splenocytes from *Ft* LVS-infected mice depends on MyD88 but not TLRs 2, 4, or 9.**
C57BL/6 and the indicated KO mice were infected with $10^5$ LVS i.d. After four days, mice were sacrificed, spleens were disrupted, and appropriate dilutions were plated on MH plates for bacterial counts (A). Values shown are the mean number of CFU/spleen ± SD of viable bacteria from three independent experiments. From the same groups, splenocytes were prepared and cultured *in vitro* for three days, and supernatants were then collected and analyzed for IFN-γ production by ELISA (B). Values shown are the mean concentration ± SD of from two independent experiments. Brackets indicate significant differences between each pair of bracketed groups ($P < 0.05$).

(Fig 2). Among the CD11c⁺ cells, we readily discriminated two groups: we classified CD11c⁺ MHCII⁺ B220⁻ Gr1⁻ CD11b⁺ NK1.1⁻ cells as conventional DC, and we found a second group (CD11c⁺ MHCII⁺/⁻ Gr1⁻ CD11b⁺/⁻ NK1.1⁺) that includes a subpopulation identified as NK-DC, also known as IKDC [49]. These cells express both NK and DC markers, and NK-DC are major producers of IFN-γ during mouse infections with *L. monocytogenes* [50]. Others have classified these cells as a natural killer subpopulation because of their functions and development [51]. Our data may support the latter interpretation, since the time course of IFN-γ production of the cells expressing both NK1.1 and CD11c markers is more similar to that of

**Table 2. Dendritic cells and neutrophils from MyD88 KO mice are highly infected with LVS but produce little IFN-γ.**

| | | | MyD88 KO | TLR2 KO |
|---|---|---|---|---|
| | | | **LVS-infected KO mice** | |
| | | | **MyD88 KO** | **TLR2 KO** |
| **Dendritic cells** | DNA | Tul4 | 2,368,944[a] | 20,894 |
| | | 23kDa | 2,932,531 | 31,553 |
| | RNA | IFN-γ | 8.8[b] | 467 |
| | | IL-1α | 1.3 | 2.4 |
| | | IL-12b | 5.3 | 1.0 |
| | | TNF-α | 0.7 | 1.0 |
| **Neutrophils** | DNA | Tul4 | 13,877,441 | 19,253 |
| | | 23kDa | 23,459,256 | 26,896 |
| | RNA | IFN-γ | 6.1 | 243 |
| | | IL-1α | 146 | 233 |
| | | IL-12b | 103 | 284 |
| | | TNF-α | 20.1 | 12.8 |

DNA and mRNA from sorted cells were prepared and used to amplify *Francisella* DNA and host cell cytokine gene expression, respectively, as described in Table 1.

[a] DNA data indicate GE per ~100 ng total DNA and DNA data are averages of two independent experiments.

[b] RNA data indicate fold change compared to cells from uninfected mice and are representative of two independent experiments.

NK cells (Fig 3A and 3B) than that of DC (Fig 3C). Nonetheless, when we depleted Rag1 KO mice of NK cells and then enriched CD11c$^+$ splenocytes four days after infection, amounts of IFN-γ produced by total DC were higher than that typically produced by total splenocytes (Fig 4) [6]. This result confirms that non-B, non-T, non-NK CD11c$^+$ cells are a substantial source of IFN-γ during innate immune responses to LVS.

Like many intracellular bacteria, macrophages represent the major target cells for intracellular infection and replication of *Ft* [21]. However, *in vitro* evidence indicates that DC can also be infected with *Ft*, and limited *in vivo* evidence suggests that *Francisella* infects DC in the lungs after respiratory infection [52]. To evaluate whether *Ft* also internalize into DC during systemic *in vivo* infection, we infected mice with either GFP- or mCherry-labeled LVS isolates that facilitated microscopy analyses (Figs 1, 5 and 6). The sensitivity of bacterial detection was limited by a combination of relatively low levels of bacteria (1 x 10$^4$–1 x 10$^5$ CFU) in the spleens of LVS-infected C57BL/6 mice, and relatively low levels of GFP or mCherry signals. A similar limitation of sensitivity was observed during flow cytometric analyses, and neither technique allowed for a satisfactory quantification of the proportion or numbers of LVS-infected DC in total cell populations from wild type mice. To improve the sensitivity of detection and convincingly demonstrate DC infection, we sorted myeloid cells after depleting B and T cells from splenocytes. DC and neutrophils could be well separated (S1 Fig) and proved suitable for LVS quantification and cytokine expression by PCR. Results were consistent with the interpretation that not only splenic DC, but also neutrophil populations were directly infected with LVS. Of note, we also attempted to sort NK cells and macrophages, but these efforts resulted in unsatisfactory cell purity and yield. Most importantly, sorting approaches demonstrated that highly purified splenic DC populations from WT mice produced high amounts of IFN-γ, as well as IL-1α.

Finally, we evaluated whether LVS-infected DC actively produced IFN-γ. The concurrent use of Rag1 KO mice and of a relatively bright GFP-LVS increased sensitivity and allowed direct visualization of individual CD11c$^+$ GFP$^+$ IFN-γ$^+$ cells using confocal microscopy (Fig 6). However, this approach also did not permit comprehensive quantification. Nonetheless,

the results demonstrated the presence of a substantial number of CD11c$^+$ cells in spleens that were simultaneously expressing IFN-γ while being infected with LVS, despite apparent disruption of infected cells (Fig 5). This visual result is consistent with a recent report indicating that cells which are in the process of dying by necroptosis, as reflected by loss of membrane integrity, apparently continue to transcribe and translate cytokine genes [53].

Signal pathways leading to the expression and production of IFN-γ are typically activated following TLR engagement. Therefore, we further investigated cytokine production by DC in MyD88 and selected TLR KO mice in response to LVS infection. We focused on mice lacking TLRs 2, 4, 9, and MyD88 because of previous studies implicating these receptors, particularly TLR2, in responses to *Francisella* [26, 27, 54–56]. Moreover, MyD88 signaling is required to limit bacterial burdens and prolong survival during pulmonary infection by virulent *Ft* SchuS4 [57]. In TLR 2, 4, or 9 KO mice, the relative amounts of IFN-γ gene expression and protein production correlated with bacterial burdens in spleens (Fig 7 and S3 Fig), suggesting that immune cells react to higher bacterial loads by producing more IFN-γ even in the face of TLR deficiencies. In contrast, cells from LVS-infected MyD88 KO mice produced low amounts of IFN-γ despite very high bacterial burdens that lead to death within 4–7 days. The observations of the relationships between bacterial burdens and IFN-γ secretion were confirmed by the analyses of DC and neutrophils sorted from splenocytes of MyD88 and TLR2 KO mice (Table 2). Higher amounts of LVS and IFN-γ gene expression were quantified in cells from TLR2 KO mice which had higher bacterial burdens, while lack of MyD88 facilitated intracellular infection but little IFN-γ production. A similar pattern was observed in purified neutrophils obtained from these mice. In contrast to IFN-γ, the expression patterns of the other cytokines we tested were similar between WT and KO mice.

Taken together, these studies confirm the critical role of IFN-γ and MyD88 in mediating innate immune responses to *Francisella* LVS infection, particularly IFN-γ production by DC in LVS-infected mice, and the importance of the final steps of the pathway that lead to IFN-γ production. In contrast, TLR2, TLR4, and TLR9 play relatively minor roles with minimal or no impact on IFN-γ production by splenocytes, including DC, and the response of the respective KO mice to LVS infection is only modestly compromised. None of these receptors appear to explain the susceptibility and poor IFN-γ production phenotype of MyD88 KO mice. IL-18 KO mice exhibit some defects in responses to *Francisella* [37, 58], but overall the MyD88-linked pattern recognition molecule critical to optimal responses to LVS by DC and other immune effector cells awaits future discovery.

## Supporting information

**S1 Fig. Gating strategy to evaluate *in vivo* NK depletion.** Rag1 KO mice were depleted *in vivo* of NK1.1$^+$ cells and then infected with $10^5$ LVS i.d, Splenocytes derived from these mice were enriched *in vitro* for CD11c$^+$ cells using magnetic beads, and the resulting cells were analyzed by flow cytometry. After exclusion of fragments, aggregates and dead cells, CD45$^+$ cells were gated for NK1.1$^+$ cells (Panel A). Alternatively, CD45$^+$ cells were excluded of Gr1$^+$ CD11b$^+$ cells and then gated for CD11c$^+$ cells (Panel B). Panels C and D show total splenocytes from Rag1 KO, not depleted of NK and not purified of CD11c.
(TIF)

**S2 Fig. Gating strategy used to sort DC and neutrophils.** C57BL/6 mice were infected with $10^5$ LVS i.d. Splenocytes from naïve and LVS-infected mice were depleted of B and T cells by magnetic beads and stained for flow cytometry. After exclusion of fragments, aggregates, and dead cells, conventional DC were sorted using CD11c and MHCII markers and cells within the upper right blue quadrant collected (A). To sort neutrophils, CD11c$^-$ MHCII$^-$ cells were

subsequently gated for CD11b$^+$ Ly6G$^+$ and cells within the upper right red quadrant were collected (B). RNA and DNA were purified from sorted cells and used for qRT-PCR (see Table 1). Data are from one independent experiment representative of three independent experiments of similar design and outcome. A similar strategy was used to sort cells from KO mice. (TIF)

**S3 Fig. IFN-γ gene expression correlates with protein production in splenocytes from LVS-infected TLR KO mice.** The indicated mice were infected with 10$^5$ LVS i.d. After four days, mice were euthanized and gene expression of IFN-γ was determined from the harvested splenocytes by qRT-PCR. Values shown are the mean Δct ± SD derived from three individual mice, multiplied by 1000 for ease of presentation. $^*$ and $^\wedge$ indicate significant differences ($P < 0.05$) between groups. (TIF)

## Acknowledgments

We would like to thank Dr. Thomas Kawula for providing GFP expression plasmid, Dr. Heather Powell and Dr. Bernard Arulanandam for providing mCherry-LVS, Dr. Tod Merkel for providing KO mice, and Mark Kukuruga for his assistance with the flow cytometry. We also thank Dr. Manuel Osorio and Dr. Mustafa Akkoyunlu for insightful discussion and thoughtful reviews of the manuscript.

## Author Contributions

**Conceptualization:** Roberto De Pascalis, Kazuyo Takeda, Adovi Akue, Karen L. Elkins.

**Data curation:** Roberto De Pascalis, Karen L. Elkins.

**Formal analysis:** Roberto De Pascalis, Lara Mittereder.

**Investigation:** Roberto De Pascalis, Amy P. Rossi, Lara Mittereder, Kazuyo Takeda, Adovi Akue, Sherry L. Kurtz.

**Supervision:** Roberto De Pascalis, Karen L. Elkins.

**Writing – original draft:** Roberto De Pascalis.

**Writing – review & editing:** Roberto De Pascalis, Amy P. Rossi, Lara Mittereder, Kazuyo Takeda, Karen L. Elkins.

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
