## [Decision Letter · Decision Letter 0]

5 Jun 2020

PONE-D-20-10838

Production of IFN-γ by splenic dendritic cells during innate immune responses against Francisella tularensis LVS depends on MyD88, but not TLR2, TLR4, or TLR9

PLOS ONE

Dear Dr. De Pascalis,

Thank you for submitting your manuscript to PLOS ONE. After careful consideration, we feel that it has merit but does not fully meet PLOS ONE’s publication criteria as it currently stands. Therefore, we invite you to submit a revised version of the manuscript that addresses the points raised during the review process.

Both expert reviewers appreciate this study and indicate that the new information contributes significantly  to the accumulated body of evidence.   However, reviewer 1 raised a concern regarding interpretation of results in one of the data graphs, and I request you to address that.  Additionally, both reviewers indicated a number of minor issues that need to be addressed in the manuscript.

We look forward to receiving your revised manuscript.

Kind regards,

Ashlesh K Murthy, M.D., Ph.D.

Academic Editor

PLOS ONE

Journal Requirements:

2. In your Methods section, please provide additional information on the animal research and ensure you have included details on : (1) methods of sacrifice (2) methods of anesthesia and/or analgesia, if relevant.

Reviewers' comments:

Reviewer's Responses to Questions

**Comments to the Author**

1. Is the manuscript technically sound, and do the data support the conclusions?

Reviewer #1: Partly

Reviewer #2: Yes

2. Has the statistical analysis been performed appropriately and rigorously? 

Reviewer #1: Yes

Reviewer #2: Yes

3. Have the authors made all data underlying the findings in their manuscript fully available?

Reviewer #1: Yes

Reviewer #2: Yes

4. Is the manuscript presented in an intelligible fashion and written in standard English?

Reviewer #1: Yes

Reviewer #2: Yes

5. Review Comments to the Author

Reviewer #1: In this paper, the authors identify dendritic cells are one of the major IFN-γ producing innate immune cells in the spleen of Francisella tularensis LVS-infected mice. The authors further demonstrated that LVS-induced IFN-γ production in DC is MyD88 dependent. These data provide insight into a previously unrecognized role of DC in innate defense against primary Ft infection by MyD88-associated IFN-γ expression. The data are well presented; however, there are some concerns to be addressed.

Major:

Comparative assessment of IFN-γ secretion by total splenocytes and NK-depleted CD11c enriched cells in Fig. 4 study is questionable. This comparison involves two variables, NK1.1 depletion and CD11c enrichment, thus no scientific conclusion can be drawn for the observed differences in cytokine level between these two samples. Assessment should be made between NK1.1 depleted and non-depleted total splenocytes or between CD11c cells enriched from anti-NK1.1 mAb treated and untreated spleen.

Minor:

1. In Fig. 3, is the y-axis a linear scale? Labeled numbers do not seem to fit the scale. Does each dot represent cell counts from a single spleen, pooled spleens or average of multiple spleen samples at each time point?

2. Is there any difference in IFN-γ producing cell population between LVS infected WT and Rag1 KO mice (lines 260-262)?

3. Lines 379-381 should be referred to S3 Fig.

4. Table 1, line 347 (and Table 2). The statement “*DNA data indicate CFU per ~100 ng DNA” is quite confusion. The bacterial number is calculated based on GE, which may not correlate with viable bacterial “CFU”. The ~100 ng “DNA” should be clearly defined and not to be confused as LVS DNA. If the mouse DNAs make up the most of the 100 ng DNA, how many mouse cells (GE) will that be?

5. Line 439. The “40%” should be clarified as in the spleen of day 4 LVS-infected mice.

Reviewer #2: The manuscript by Pascalis et al demonstrates that Ft LVS can infect DC and induce IFN-g in the spleens of mice following i.d infection. Additionally, they show that IFN-g production by DC is dependent on Myd88 but not TLR2/4/9. These results are interesting and critical for our understanding of vaccine-induced immunity to Tularemia. The manuscript is well written and easy to understand with adequate statistical analysis.

Minor concerns:

This is a suggestion as it may be beyond the scope of the manuscript. It would have been useful to determine the fraction of IFN-g positive cells that are also positive for LVS and compare that to IFN-g positive DCs that are negative for LVS. Authors do indicate that they failed to differentiate such populations due to technical difficulties. An alternative way of determining this may be to utilize in vitro cultured DCs and infecting them with LVS and performing the assay over a period of time. While I acknowledge this may not reflect the in vivo effect, data from such experiments allow for further molecular characterizing of the mechanism underlying the production of IFN-g.

While the discussion section describes the results in the context of previous literature, a further discussion of the following points will benefit the readers. 1) How do these results compare to i.n. infection, particularly that of sub lethal i.n. infection. 2) How do these results compare to that of lethal i.d. infection.

6. PLOS authors have the option to publish the peer review history of their article (what does this mean?). If published, this will include your full peer review and any attached files.

Reviewer #1: No

Reviewer #2: No

---

## [Author Response · Author response to Decision Letter 0]

24 Jun 2020

Manuscript Ref: PONE-D-20-10838

Title: Production of IFN-γ by splenic dendritic cells during innate immune responses against Francisella tularensis LVS depends on MyD88, but not TLR2, TLR4, or TLR9

Journal: PLOS ONE

We would like to thank the reviewers for the thoughtful comments, and we acknowledge the additional journal requirements. Below, we include the original comments of the editor and reviewers in italics and provide a response to each individual comment. Because line numbers have changed, the resulting changes to the manuscript have been referred by the line numbers of the revised version only.

We apologize for overlooking this aspect. Changes have been made throughout the manuscript, including file naming as per PLOS ONE’s style requirements.

2 In your Methods section, please provide additional information on the animal research and ensure you have included details on: (1) methods of sacrifice (2) methods of anesthesia and/or analgesia, if relevant.

The method of sacrifice has been added as per journal requirement (lines 93 – 95). Although animals were not subjected to anesthesia or analgesia, a sentence has been added to address this point (lines 95 – 96).

Reviewer #1

Major:

Comparative assessment of IFN-γ secretion by total splenocytes and NK-depleted CD11c enriched cells in Fig. 4 study is questionable. This comparison involves two variables, NK1.1 depletion and CD11c enrichment, thus no scientific conclusion can be drawn for the observed differences in cytokine level between these two samples. Assessment should be made between NK1.1 depleted and non-depleted total splenocytes or between CD11c cells enriched from anti-NK1.1 mAb treated and untreated spleen.

We apologize for the misleading description of the figure, which led the reviewer to focus on comparing results between the two groups. This was not our intent, and we agree that the studies proposed by the reviewer would be needed to support direct comparisons. Here, however, the goal of this experiment was simply to evaluate the IFN-γ production by highly enriched splenic DC (i.e., cells obtained from Rag1 KO mice that were depleted in vivo of NK1.1+ cells and enriched in vitro for CD11c+ cells). We found that IFN-γ production by highly enriched CD11c+ cells was substantial, to such an extent that it was very unlikely to be due to non- CD11c+ cells. To provide context for the amount, we showed what typically is obtained from total Rag1 KO splenocytes alongside. To avoid confusion, we modified the layout of the figure and its description. The text has been modified to capture these changes (lines 303 – 313; 318-320).

Minor:

1 In Fig. 3, is the y-axis a linear scale? Labeled numbers do not seem to fit the scale. Does each dot represent cell counts from a single spleen, pooled spleens or average of multiple spleen samples at each time point?

The reviewer is correct: the Y-axis is a linear scale. The range for numbers of each subpopulation is different. Therefore, to graphically emphasize the pattern of IFN-γ-producing cells, the scale for each panel is different. 

Each dot represents cell counts from three pooled spleens. We apologize for not including this information, and we added a sentence to address this point (lines 300 – 301).

2 Is there any difference in IFN-γ producing cell population between LVS infected WT and Rag1 KO mice (lines 260-262)?

In earlier studies (De Pascalis et al., I&I, 2008), we demonstrated that the numbers of IFN-γ producing DC, NK and NK-DC were comparable between LVS-infected WT and Rag1 KO mice. Differences were related to the lack of IFN-γ producing T and NK T cells. We have added a sentence to address the reviewer’s comment and clarify this point in the text (lines 270 – 272).

3. Lines 379-381 should be referred to S3 Fig.

We apologize for the mistake. We corrected this typographical error (line 397).

4. Table 1, line 347 (and Table 2). The statement “*DNA data indicate CFU per ~100 ng DNA” is quite confusion. The bacterial number is calculated based on GE, which may not correlate with viable bacterial “CFU”. The ~100 ng “DNA” should be clearly defined and not to be confused as LVS DNA. If the mouse DNAs make up the most of the 100 ng DNA, how many mouse cells (GE) will that be?

We apologize for the confusion. The reviewer is correct that GE may not correlate with CFU. We have therefore modified the language throughout the manuscript to better describe GE measurements (lines 205 – 207; 361 – 365; 427 – 428). In addition, we specifically pointed out that 100 ng DNA includes murine and bacterial DNA (line 201 and 363). To answer the reviewer’s question about how many cells are in 100 ng DNA, if the size of mouse DNA is 1.8x1012 daltons, where 1 dalton is 1.67x10-24 g, we estimate that the mouse genome would be ~ 3x10-12 g or 3 pg. Therefore, 100 ng of mouse DNA derives from ~ 33000 cells. As the reviewer pointed out, the majority of this 100 ng derives from mouse cells. 

5. Line 439. The “40%” should be clarified as in the spleen of day 4 LVS-infected mice

We agree and appreciate the reviewer’s suggestion for clarification. The sentence has been modified (line 456).

Reviewer #2:

Minor concerns:

1. This is a suggestion as it may be beyond the scope of the manuscript. It would have been useful to determine the fraction of IFN-g positive cells that are also positive for LVS and compare that to IFN-g positive DCs that are negative for LVS. Authors do indicate that they failed to differentiate such populations due to technical difficulties. An alternative way of determining this may be to utilize in vitro cultured DCs and infecting them with LVS and performing the assay over a period of time. While I acknowledge this may not reflect the in vivo effect, data from such experiments allow for further molecular characterizing of the mechanism underlying the production of IFN-γ.

We appreciate the reviewer’s suggestion, although we do consider this to be beyond the scope of this manuscript. This is in part because we explored this approach briefly and encountered several limitations. We found that bone marrow-derived DCs were inherently more difficult to infect with Francisella LVS compared to bone marrow-derived macrophages, and the degree of infection depended on culture conditions and multiplicity of infection. These variables limited a satisfying, direct evaluation of LVS-infected DCs producing IFN-γ. In addition, as the reviewer pointed out, the in vitro findings may not apply to the in vivo effects. We therefore did not pursue this line of investigation further.

2. While the discussion section describes the results in the context of previous literature, a further discussion of the following points will benefit the readers. 1) How do these results compare to i.n. infection, particularly that of sub lethal i.n. infection. 2) How do these results compare to that of lethal i.d. infection.

We thank the reviewer for these useful suggestions. Sentences and references have been added to capture these points (lines 64 – 66; 68 – 70; and 76).

In the course of revisions, other minor text edits have also been incorporated into the manuscript. Collectively, we believe these revisions have resulted in a substantially improved paper. We appreciate your consideration of the revised manuscript and look forward to your reply.

Sincerely,

Roberto De Pascalis and Karen L. Elkins, Ph.D.

Laboratory of Mucosal Pathogens and Cellular Immunology

Division of Bacterial, Parasitic, and Allergenic Products

Office of Vaccines Research and Review

Center for Biologics Evaluation and Research, U.S. FDA

10903 New Hampshire Avenue

Building 52/72, Room 5328 or 5330

Silver Spring, MD 20903

Telephone: 240.402.9465 (RDP) or 240.402.9507 (KLE)

FAX: 301.595.1235

E-mail: roberto.depascalis@fda.hhs.gov or karen.elkins@fda.hhs.gov

---

## [Decision Letter · Decision Letter 1]

20 Jul 2020

Production of IFN-γ by splenic dendritic cells during innate immune responses against Francisella tularensis LVS depends on MyD88, but not TLR2, TLR4, or TLR9

PONE-D-20-10838R1

Dear Dr. De Pascalis,

We’re pleased to inform you that your manuscript has been judged scientifically suitable for publication and will be formally accepted for publication once it meets all outstanding technical requirements.

Kind regards,

Ashlesh K Murthy, M.D., Ph.D.

Academic Editor

PLOS ONE

Additional Editor Comments (optional):

Reviewers' comments:

Reviewer's Responses to Questions

**Comments to the Author**

1. If the authors have adequately addressed your comments raised in a previous round of review and you feel that this manuscript is now acceptable for publication, you may indicate that here to bypass the “Comments to the Author” section, enter your conflict of interest statement in the “Confidential to Editor” section, and submit your "Accept" recommendation.

Reviewer #1: All comments have been addressed

2. Is the manuscript technically sound, and do the data support the conclusions?

Reviewer #1: Yes

3. Has the statistical analysis been performed appropriately and rigorously? 

Reviewer #1: Yes

4. Have the authors made all data underlying the findings in their manuscript fully available?

Reviewer #1: Yes

5. Is the manuscript presented in an intelligible fashion and written in standard English?

Reviewer #1: Yes

6. Review Comments to the Author

Reviewer #1: (No Response)

7. PLOS authors have the option to publish the peer review history of their article (what does this mean?). If published, this will include your full peer review and any attached files.

Reviewer #1: No

---

## [Editor Report · Acceptance letter]

23 Jul 2020

PONE-D-20-10838R1 

Production of IFN-γ by splenic dendritic cells during innate immune responses against Francisella tularensis LVS depends on MyD88, but not TLR2, TLR4, or TLR9 

Dear Dr. De Pascalis:

I'm pleased to inform you that your manuscript has been deemed suitable for publication in PLOS ONE. Congratulations! Your manuscript is now with our production department. 

Kind regards, 

on behalf of

Dr Ashlesh K Murthy 

Academic Editor

PLOS ONE